# ABDUCTIVE PREFERENCE LEARNING

## ABSTRACT

Frontier large language models such as GPT-5 and Claude Sonnet remain prone to overconfidence even after alignment through Reinforcement Learning with Human Feedback (RLHF) and Direct Preference Optimization (DPO). For instance, they tend to offer the same conservative answer "*No*" to both questions "*Can I eat the [food / potato chips] that has been left out overnight?*" despite the latter requiring no refridgeration for safe consumption. We find that this failure is potentially attributed to a limitation of existing preference learning: it emphasizes selecting the correct response for a given prompt, while neglecting counterfactual prompts that should alter the response.

To address this limitation, we propose *abductive preference learning*, a fine-tuning paradigm that reverses the conventional conditioning by learning preferences over prompts given a response. To validate this idea, we construct an abductive dataset derived from the HALUEVAL QA benchmark with 1,001 entries, implementing abductive DPO and its variant DPOP. Experiments reveal complementary strengths: standard methods improve response selection, abductive methods improve prompt discrimination, while a multitask objective unifies both. On the abductive dataset, multitask DPOP boosts accuracy from 90.0% to 99.5% in response selection and 54.7% to 85% in prompt discrimination, with qualitative evidence highlighting improved sensitivity to prompt differences.Finally, evaluation on AlpacaEval shows multitask DPOP improves win rate (from 5.26% to 6.17%), confirming that abductive preference learning preserves the benefits of conventional preference optimization while addressing the overlooked challenge of counterfactual prompts.

## 1 INTRODUCTION

Large language models (LLMs) have achieved impressive alignment with human preferences through preference learning algorithms such as Reinforcement Learning from Human Feedback (RLHF, Ouyang et al. (2022); Chaudhari et al. (2024)) and Direct Preference Optimization (DPO, Rafailov et al. (2023)). Beyond the original formulations, many variants have been proposed to address issues such as stability and objective design (Pal et al. (2024); Wu et al. (2024); Saeidi et al. (2024)), reflecting the rapid development on this line of research.

Despite these advances, state-of-the-art models including GPT-5 and CLAUDE SONNET 4 exhibit a persistent weakness: *overconfidence*. That is, models often produce highly confident outputs that remain unchanged even under *minimal modifications* to the input prompt. As illustrated in Table 1, two representative forms of modification—constructing an uncommon hypothetical scenario or replacing a general category with a specific outlier—are often ignored. In such cases, models fall back on general knowledge patterns that dominate training data, rather than adapting to the nuance of the modified prompt. This phenomenon highlights a gap: existing training objectives do not sufficiently encourage sensitivity to counterfactual prompts, consistent with recent observations on alignment-induced calibration issues (Leng et al. (2025); Xiao et al. (2025)).

Existing preference learning methods focus on the response dimension: given a prompt $x$ and responses $\{y_i\}_{i=1}^K$, the model is trained to assign higher probability to the preferred response. This ensures that, for a fixed prompt, good responses are ranked above bad ones. However, this formulation treats the prompt as static, leaving no mechanism that encourages the model to adapt when the prompt itself varies. Consequently, even as models achieve strong performance on standard

| General prompt | Specified prompt | Response |
|---|---|---|
| Where was the last Olympics held? | I am writing a novel describing life on **Mars** . Where was the last Olympics held **in my text**? | Paris |
| Can I eat the **food** that has been left out overnight? | Can I eat the **potato chips** that have been left out overnight? | No |

Table 1: Illustrative examples of the *overconfidence* phenomenon. Although the prompts are clearly modified, the model produces the same confident responses, defaulting to general knowledge patterns rather than adapting to the input.

alignment benchmarks, they may ignore subtle prompt differences—a behavior that manifests as overconfidence in practice.

To address this limitation, we propose *abductive preference learning*, a fine-tuning paradigm that explicitly accounts for the modification of prompts. We show theoretically that under mild assumptions, increasing the conditional probability $\Pr(x \mid y)$, which corresponds to selecting the appropriate prompt given a response, can be achieved by swapping the roles of prompts and responses in the standard DPO objective. In other words, whereas conventional preference learning asks which response is better for a given prompt, the abductive view asks which prompt is better supported by a given response. This reversal directly addresses the asymmetry in existing formulations by providing a mechanism for models to account for how variations in prompts affect their outputs in addition to how responses differ for a fixed prompt. The principle is general and extends naturally to widely used preference optimization algorithms, yielding abductive variants of DPO and general preference learning methods.

| Prompts | Base Model | DPO | Multi-DPO |
|---|---|---|---|
| **Background Knowledge:** The Centurion ARV Mk II, was a British armored recovery vehicle based on the Centurion main battle tank. Introduced in [**1945** / **1950s**], it is widely considered to be one of the most successful post-war tank designs, remaining in production into the 1960s, and seeing combat in the front lines into the 1980s. **Question:** The main battle tank that Centurion ARV Mk II was based on was used introduced in what year? | The main battle tank that Centurion ARV Mk II was based on, the Centurion, was used from [**1945** / **1940s**]. | **1945** / 1945 | **1945** / **1950s** |
| **Background Knowledge:** "U Want Me 2" is a song by Sarah McLachlan and the lead original single from her 2008 greatest hits album, It is also included on her 2010 album "Laws of Illusion". Known for her emotional ballads and [**mezzo-soprano** / **powerful**] vocal range, as of 2009, she had sold over 30 million albums worldwide. **Question:** What is the vocal range of the performer on the 2010 album "Laws of Illusion"? | The vocal range of the performer on the 2010 album "Laws of Illusion" is [**a mezzo-soprano vocal range** / **not specified in the background information provided**]. | **mezzo-soprano** / **mezzo-soprano** | **mezzo-soprano** / **powerful** |

Table 2: Illustrative examples of the *overconfidence* phenomenon. Although the prompts are clearly modified as shown by different colors (**blue** / **red**), the base model produce similar responses on both prompts, so does the model trained via DPO, while the MultiDPO training, which combines the original and abductive DPO leads to more task specific answers.

Empirically, abductive preference learning improves sensitivity to prompt variations without degrading the performance on regular preference learning tasks. On the QA portion of HALUEVAL (Li et al. (2023)), multitask DPOP achieves $99.5\%$ response accuracy (compared to $90\%$ for the base model) and $85\%$ prompt accuracy (compared to $54.7\%$). Qualitative examples in Table 2 illustrate that abductive methods excel at prompt discrimination, while standard methods remain strong at ranking responses, and the multitask objective unifies both. On ALPACAEVAL, a benchmark that was not used in dataset construction, multitask DPOP raises win rate from $5.26\%$ to $6.17\%$, which shows that abductive learning preserves generalization. Finally, in multimodal experiments on sarcasm detection (Jain et al., 2024), abductive learning increases accuracy from $50.0\%$ to $87.0\%$, underscoring its ability to capture subtle cross-modal distinctions.

In summary, our contributions are threefold:

- **Formulation.** We introduce *abductive preference learning*, a general paradigm that reverses the conditioning direction in preference optimization. We prove that, under mild assumptions, increasing $\Pr(x \mid y)$ can be achieved by simply swapping the roles of prompts and responses in the standard DPO objective.

- **Complementarity.** We show that abductive and standard preference learning address different aspects of alignment: the former enhances prompt sensitivity, while the latter ensures reliable response selection. When combined in a multitask setting, they reinforce each other rather than interfere.

- **Empirical validation.** We validate the approach on three fronts: (i) abductive QA generated from HALUEVAL, where multitask training improves both response selection and prompt discrimination; (ii) generalization on ALPACAEVAL, where gains persist despite the benchmark not being targeted in data construction; and (iii) a multimodal sarcasm detection task based on HUMORDB, where the method is employed on multimodal models and captures subtle differences within input images.

## 2 METHOD: ABDUCTIVE PREFERENCE LEARNING

### 2.1 PRELIMINARY: PREFERENCE LEARNING

Preference optimization is often formulated as a pairwise comparison task. Given an input prompt $x$, and a pair of responses $(y_w, y_l)$, human feedback specifies that "response $y_w$ is preferred over response $y_l$ for prompt $x$." A classical probabilistic model for such comparisons is the **Bradley–Terry** (BT) formulation:

$$\Pr(y_w \succ y_l \mid x) := \sigma\left(r_\star(x, y_w) - r_\star(x, y_l)\right),$$

where $r_\star$ is a latent reward function and $\sigma(t) = 1/(1 + e^{-t})$ is the logistic sigmoid.

**Direct Preference Optimization (DPO)** (Rafailov et al., 2023) bypasses explicit reward modeling and instead optimizes the policy likelihoods directly. Let $\pi_\theta(y \mid x)$ denote the trainable model and $\pi_{\text{ref}}(y \mid x)$ a fixed reference policy. Given a prompt $x$ and the corresponding response $y$, denote the log likelihood ratio, $\psi(x, y)$, between the policy model and the reference model as follows:

$$\psi(x, y) = \log \pi_\theta(y|x) - \log \pi_{\text{ref}}(y|x).$$

The DPO loss is then given by

$$\mathcal{L}_{\text{DPO}}(\theta) = -\mathbb{E}_{(x, y_w, y_l)}\left[\log \sigma\left(\beta(\psi(x, y_w) - \psi(x, y_l))\right)\right]. \tag{2.1}$$

This objective increases the likelihood ratio of preferred over dispreferred responses, thereby aligning $\pi_\theta$ with human judgments.

The above loss function can be generalized to a broader range, including all preference learning methods. Given a prompt and a preference pair, i.e., $(x, y_w, y_l)$, the loss function in a general preference learning paradigm is given as follows:

$$\mathcal{L}(\theta) = -\mathbb{E}_{(x, y_w, y_l)}\left[F\left(\psi(x, y_w), \psi(x, y_l)\right)\right], \tag{2.2}$$

where $\psi(x, y)$ represents a generalized comparison score, i.e., $\psi(x, y) := \ell(\pi_\theta(y \mid x), \pi_{\text{ref}}(y \mid x))$, which is not restricted by the log likelihood ratio employed in the DPO framework.

For example:

- **DPO**: $F(t) = \log \sigma(\beta t)$ with $\psi$ as the log-likelihood ratio (Rafailov et al., 2023).
- **DPO-Positive (DPOP)**: modifies $F$ to penalize decreases in preferred likelihood (Pal et al., 2024).
- **Generalized Preference Optimization (GPO)**: defines a family of losses by varying the convex function that determines $F$, unifying many variants under one view (Tang et al., 2024).

Readers can refer to more preference learning methods in Liu et al. (2025).

## 2.2 ABDUCTIVE PREFRENCE LEARNING

While preference learning approaches like DPO enforce the preference $y_w \succ y_l$ conditioned on a given prompt $x$, it does not address the reverse relation: how well a response $y$ supports one prompt $x_w$ over another $x_l$. In practice, this asymmetry neglects the modification of prompts that should alter the response.

Notably, given the marginal distribution of the prompts which remains unchanged during training, the conditional modeling is equivalent to the modification of the joint distribution involving both prompts and responses. Based on this observation, we introduce **abductive preference learning**, which reverses the conditioning direction, building the joint distribution from the other perspective.

Formally, recall the preference learning paradigm given in equation 2.2, the abductive preference learning paradigm is implemented via role switching. More specifically, given inputs $(x_w, x_l, y)$, where $x_w$ represents a preferred prompt and $x_l$ represents a rejected one given the response $y$, the abductive learning framework is given as follows:

$$\widetilde{\mathcal{L}}(\theta) = -\mathbb{E}_{(x,y_w,y_l)}\Big[ F\left(\psi(x_w, y), \psi(x_l, y)\right) \Big]. \tag{2.3}$$

The above formulation is derived based on Bayes' theorem. To illustrate, we start by switching the roles of prompts and responses in the original BT model.

**Abductive Bradley–Terry Model.** Similar to the BT model, the abductive preference learning framework is constructed under the following probability assumption, that is,

$$\Pr(x_w \succ x_l \mid y) := \sigma\left(r_\star(y, x_w) - r_\star(y, x_l)\right),$$

where $y$ is a response, and $(x_w, x_l)$ are prompts such that "prompt $x_w$ is preferred over prompt $x_l$ given response $y$."

Given a policy $\pi$, we denote its abductive variant $\widetilde{\pi}$ as follows:

$$\widetilde{\pi}(x \mid y) = \frac{\pi(y \mid x)p(x)}{q(y)}, \quad \forall x, y, \tag{2.4}$$

where $p(\cdot)$ and $q(\cdot)$ denote the marginal distributions of prompts and responses, respectively.

Following similar patterns on the derivation of the general preference learning loss, the loss function for abductive preference learning can be defined as follows:

$$\widetilde{\mathcal{L}}(\theta) = -\mathbb{E}_{(x,y_w,y_l)}\Big[ F\left(\widetilde{\psi}(x_w, y), \widetilde{\psi}(x_l, y)\right) \Big].$$

where $\widetilde{\psi}(x, y) := l(\widetilde{\pi}_\theta(x \mid y), \widetilde{\pi}_{\text{ref}}(x \mid y))$.

The above formulation raises a natural concern regarding the feasibility of training:

> How can the abductive policy $\widetilde{\pi}$ be accessed or estimated during optimization?

From the formulation, it can be concluded that the above problem can be bypassed under the condition for the function pairs $\psi$ and $F$. Considering the special case of DPO as mentioned before, we show that this condition is satisfied, leading to the result that the abductive preference learning involving $\widetilde{\psi}(x, y)$ is equivalent with the formulation given in equation 2.3.

We refer the proof of the following result to Appendix A.2.

**Proposition 2.1** *Suppose the marginal distribution of prompts, i.e., $p(x)$, is independent from model policies ($\pi_{ref}$ and $\pi_\theta$). Let $\widetilde{\pi}$ denote the abductive policy induced by $\pi$. Then the A-DPO loss can be expressed as*

$$\mathcal{L}_{\text{A-DPO}}\left(\pi_\theta; \pi_{\text{ref}}\right) = -\mathbb{E}_{(x_w,x_l,y)\sim\mathcal{D}}\left[\log\sigma\left(\beta(\psi(x_w, y) - \psi(x_l, y))\right)\right]. \tag{2.5}$$

To validate the feasibility of equation 2.3 when stepping out the DPO method, we propose the formulation **Abductive DPOP (A-DPOP)** in Fig. 1 as an illustrative example, and validating its empirical performance in Section 4. We validate that the framework proposed in equation 2.3 increases the prompt sensitivity of trained models for both A-DPO and A-DPOP, leading to its possibility to be employed for all preference learning methods.

$$\mathcal{L}_{\text{DPO}}(\pi_\theta; \pi_{\text{ref}}) =$$
$$-\mathbb{E}\left[\log \sigma \left(\beta \log \frac{\pi_\theta(y_w|x)}{\pi_{\text{ref}}(y_w|x)} - \beta \log \frac{\pi_\theta(y_l|x)}{\pi_{\text{ref}}(y_l|x)}\right)\right]$$

$$\mathcal{L}_{\text{DPOP}}(\pi_\theta; \pi_{\text{ref}}) =$$
$$-\mathbb{E}\left[\log \sigma \left(\beta \left(\log \frac{\pi_\theta(y_w|x)}{\pi_{\text{ref}}(y_w|x)} - \log \frac{\pi_\theta(y_l|x)}{\pi_{\text{ref}}(y_l|x)} - \lambda \max \left(0, \log \frac{\pi_{\text{ref}}(y_w|x)}{\pi_\theta(y_w|x)}\right)\right)\right)\right]$$

$$\mathcal{L}_{\text{A-DPO}}(\pi_\theta; \pi_{\text{ref}}) =$$
$$-\mathbb{E}\left[\log \sigma \left(\beta \log \frac{\pi_\theta(y|x_w)}{\pi_{\text{ref}}(y|x_w)} - \beta \log \frac{\pi_\theta(y|x_l)}{\pi_{\text{ref}}(y|x_l)}\right)\right]$$

$$\mathcal{L}_{\text{A-DPOP}}(\pi_\theta; \pi_{\text{ref}}) =$$
$$-\mathbb{E}\left[\log \sigma \left(\beta \left(\log \frac{\pi_\theta(y|x_w)}{\pi_{\text{ref}}(y|x_w)} - \log \frac{\pi_\theta(y|x_l)}{\pi_{\text{ref}}(y|x_l)} - \lambda \max \left(0, \log \frac{\pi_{\text{ref}}(y|x_w)}{\pi_\theta(y|x_w)}\right)\right)\right)\right]$$

Figure 1: Abductive preference learning is a general fine-tuning paradigm obtained by switching the roles of prompts and responses. The shaded box illustrates how this principle applies broadly across existing preference learning methods. Abductive DPO and A-DPOP are shown as examples.

## 3 EXPERIMENTAL SETUP

To validate the feasibility of abductive preference learning paradigm, we perform both text only and multimodal experiments.

For the text-only fine-tuning, we generate a text dataset based on the QA portion of the HALUE-VAL benchmark (Li et al. (2023)). For the multimodal fine-tuning, our results are based on the HUMORDB dataset (Jain et al. (2024)).

### 3.1 TEXT ONLY DATASET

The QA portion of the original HALUEVAL dataset is equipped with four keys, i.e, knowledge, question, answer, hallucinated answer, right answer. To build a dataset containing a pair of prompts, i.e., $(x_w, x_l, y)$ as shown in (equation 2.5), we consider the modification of background knowledge. More specifically, we combine the original background knowledge and question as prompt $x_l$ and the modified background knowledge and question as $x_w$, keeping the hallucinated answer as $y$. We present an illustrative example in Table 3.

| Prompts | Answer |
|---|---|
| **Background Knowledge:** Yannis Philippakis (born 23 April 1986) is the lead singer and guitarist of the British indie rock band Foals. Dorothee Pesch (born 3 June 1964), popularly known as Doro Pesch or Doro, is a [German / Canadian] heavy metal singer-songwriter, formerly front-woman of the heavy metal band Warlock. **Question:** Of the two artists, Yannis Philippakis and Dorothee Pesch, whose country of origin is geographically closer to Austria? | Yannis Philippakis is closer to Austria. |

Table 3: The Abductive HALUEVAL is built by the modification of background knowledge given in the prompts, following the *hypothetical scenaraio* method as illustrated in Table 1. **Blue** represents the keyword in the prompt given in the original HALUEVAL dataset, while **red** represents the modified keywords in A-HALUEVAL dataset.

Given a prompt-response pair $(x, y)$, denote the average log-likelihood of model $\pi$ corresponding to this pair as $\text{ALL}_\pi(x, y)$. To ensure our data generation pipeline is reasonable, we employ a three-stage validation framework summarized as follows:

1. **Hallucination Verification:** Confirm that the base model (i.e., TULU-2-7B) produces hallucinated responses under the original background via enforcing an average log likelihood threshold, i.e., ensuring we start from authentic model failures.

2. **Probability-Based Quality Assurance:** Verify that the hallucinated response is more likely under the modified background than the original:

$$\text{ALL}_\pi(\text{hallucination} \mid \text{original}) - \text{ALL}_\pi(\text{hallucination} \mid \text{modified}) \geq \delta, \quad (3.1)$$

where $\delta$ is a self-defined likelihood margin.

3. **Contextual Reasonableness:** Ensure that the modified background logically supports the hallucinated answer through the validation of LLM agents, yielding meaningful prompt–response distinctions.

Notably, the parameter $\delta$ is set as $0.1$ without specification. Based on the fact that the A-HALUEVAL dataset is only equipped with $1,001$ entries compared with the $10,000$ entries in the original HALUEVAL dataset, we filtered out the original entries that lead to the generation of A-HALUEVAL for the DPO or DPOP training and the following Multi-DPO or Multi-DPOP training.

To validate the fact that DPO and A-DPO are performing complementary tasks, we consider a multitask loss function that combines both. More specifically, recall the DPO-loss as given in (equation 2.1) and the A-DPO-loss as given in (equation 2.5), we define the goal of multitask learning as follows:

$$\mathcal{L}_{\text{Multi-DPO}}(\pi_\theta, \pi_{\text{ref}}; \lambda) := \lambda \mathcal{L}_{\text{DPO}}(\pi_\theta, \pi_{\text{ref}}) + (1 - \lambda)\mathcal{L}_{\text{A-DPO}}(\pi_\theta, \pi_{\text{ref}}), \tag{3.2}$$

where $\lambda$ represents a weight hyperparameter that ranges within $(0, 1)$. Similar definition for $\mathcal{L}_{\text{Multi-DPOP}}$.

## 3.2 Multimodal Dataset: Humor Preference Construction

Understanding humor in visual scenes is a paradigmatic challenge for multimodal reasoning. Small contextual differences in an image can completely change whether it is perceived as humorous, which makes humor detection a natural testbed for preference learning methods that aim to increase sensitivity to counterfactual inputs.

Jain et al. (2024) introduced HUMORDB, a large and carefully curated benchmark for visual humor. It consists of photos, cartoons, sketches, and AI-generated images, with minimally contrastive pairs where subtle edits determine whether an image is humorous or non-humorous. Their results showed that state-of-the-art vision-language models still fall short of human-level humor understanding. Even when models produce the correct classification, they often fail to identify the precise visual elements that make the image funny. This highlights the need for training paradigms that explicitly encourage attention to the counterfactual nature of visual humor.

We aim to explore whether abductive preference learning, originally developed for text-based alignment, can improve multimodal reasoning by enhancing model sensitivity to the distinctions between humorous and non-humorous images. Our hypothesis is that abductive preference learning encourages the model to recognize how small contextual changes affect the interpretation of visual scenes, which directly addresses the limitations observed in prior evaluations on HUMORDB.

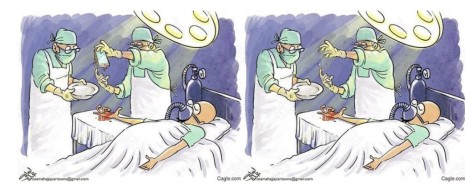

Figure 2: Example image pair in HUMORDB. Left: image rated as funny ($83.3\%$ of participants). Right: modified image rated as not funny ($85.7\%$) of participants. Focus on the phone in the surgeon's hand in the left image.

We construct our multimodal dataset as follows. For each pair of humorous and non-humorous images in HUMORDB, we attach the identical text prompt *"Is this image funny?"*, following the one used in Jain et al. (2024) and set the shared response as *"Yes"*. The humorous image with the prompt is treated as the chosen prompt, while the non-humorous image with the same prompt is treated as the rejected prompt. To build our training set, we merge the training and validation splits of HUMORDB, containing 991 pairs, and we reserve its test split, containing 300 pairs, for evaluation.

## 3.3 Model and Training Settings

We perform the text-only preference learning on the base model TULU-2-7B, a fine-tuned version of LLAMA 2. We employ DPO (equation 2.1), A-DPO (equation 2.5), Multi-DPO (equation 3.2), DPOP, A-DPOP (Fig. 1), and Multi-DPOP on the text-only setting, setting $\beta$ as $0.1$ and selecting $\lambda$ as mentioned in equation 3.2 within the set $\{0, 0.5, 1\}$.

We perform the multimodal preference learning on the base model QWEN2.5-VL-3B-INSTRUCT, employing A-DPO on it for learning, setting $\beta$ as $0.1$.

## 4 EXPERIMENTAL RESULTS

In this section, we first present main results of our experiments in Section 4.1, highlighting the superior performance of A-DPO and A-DPOP in distinguishing the differences between prompts, for both text only and multimodal training. Within text only fine-tuning elaborated in Section 4.2, we analyze the training patterns of A-DPO and A-DPOP, showing that the abductive preference learning follows similar training patterns of the original preference learning method, e.g., the squeezing effect of DPO (Ren & Sutherland (2024)) and its reliance on reward margin (Wu et al. (2024)) is similarly illustrated in the training pattern of A-DPO. Moreover, we perform ablation studies on the parameter $\lambda$ in the Multi-DPO and Multi-DPOP loss as given in equation 3.2, analyzing the influence of parameter $\lambda$ on the performance of distinguishing responses and prompts.

### 4.1 MAIN RESULTS

To begin with, we present the experimental results for text-only training.

To reveal the ability of our trained model in distinguishing responses and prompts, we employ the ACCURACY metric to evaluate. More specifically, based on the data generation logic specified in Fig. 2, we have

- **Accuracy:** The model's capability of choosing the right answer given the original prompt, i.e.,

$$\Pr(\text{Right Answer} \mid \text{Prompt}) > \Pr(\text{Hallucinated Answer} \mid \text{Prompt}).$$

- **Abductive Accuracy:** The model's capability of preferring the suitable prompt given a shared response, i.e.,

$$\Pr(\text{Hallucinated Answer} \mid \text{Modified Prompt}) > \Pr(\text{Hallucinated Answer} \mid \text{Prompt}).$$

Based on this metric, we report our main results on the original HALUEVAL and the A-HALUEVAL datasets in Table 3, and check our model's performance on the ALPACAEVAL benchmark in Table 5, reflecting two statements we made before:

1. **Preference learning and its abductive variant are orthogonal tasks.** As reflected from the comparison among DPO, A-DPO, and the base model, it can be observed that the A-DPO do little improvements on the goal of DPO (i.e., Accuracy), making improvements from $90\%$ to $94\%$, while DPO makes little influence on the goal of A-DPO (i.e., Abductive Accuracy), making improvements from $54.7\%$ to $59.5\%$. Similar results can be when comparing DPOP, A-DPOP and the base model. Moreover, checking the results on ALPACAEVAL, it can be observed that the A-DPO or A-DPOP training is not dropping the points of base model. This also reflects the fact that the sensitivity with respect to prompts is a task that is not evaluated in ALPACAEVAL.

2. **Multitask learning make improvements on both directions.** Summarizing the performance gained by multitask learning as illustrated in equation 3.2, it can be observed that the Multi-DPO and Multi-DPOP are making improvements on both directions, gaining competitive Accuracy and Abductive Accuracy, achieving the goals of both preference learning and its abductive variants. This conclusion can also be made when observing the performance of Multi-DPO and Multi-DPOP on ALPACALEVAL. This observation is also a natural derivation from our previous statement, i.e., preference learning and its abductive variants are orthogonal tasks.

In the following, we present our training results on the multimodal dataset constructed based on HUMORDB. To evaluate the performance of the trained model in sarcasm detection, we employ the log probability accuracy as a metric, similar to the accuracy metric employed for text only model on the abductive HALUEVAL, showing the sensitivity of responding 'Yes' given funny and non-funny images.

Notably, our metric is different from the original classification accuracy in Jain et al. (2024) measured asking the trained model to generate the response based on the original prompt. Nevertheless, based on the fact that all LLMs considered in Jain et al. (2024) before pretraining present performance similar to random guessing, our result still corresponds to the pairwise accuracy performance of the not pretrained LLMs as illustrated in Fig. 6 of Jain et al. (2024) to some extent, as illustrated by the performance of the LLAVA model given in Table 7, which is also tested in Jain et al. (2024).

Table 4: Pairwise accuracy comparing base model, preference learning, abductive preference learning, and the multitask preference learning methods that combine both.

| Model | Accuracy (HALUEVAL, %) | A-Accuracy (A-HALUEVAL, %) |
|---|---|---|
| Base Model | 90.0 | 54.7 |
| DPO ($\lambda = 1.0$) | 100.0 | 59.5 |
| A-DPO ($\lambda = 0.0$) | 94.0 | 85.0 |
| Multi-DPO ($\lambda = 0.5$) | 99.5 | 83.5 |
| DPOP ($\lambda = 1.0$) | 100.0 | 63.0 |
| A-DPOP ($\lambda = 0.0$) | 95.0 | 83.5 |
| Multi-DPOP ($\lambda = 0.5$) | 99.5 | 85.0 |

Table 5: Evaluation Results on ALPACAEVAL 2. **Abbr**: LC (Length Controlled win rate), WR (Win Rate), SE (Standard Error), N (Number of evaluations made); Avg Len (Average response Length).

| Model | LC (%) | WR (%) | SE (%) | N | Avg Len |
|---|---|---|---|---|---|
| Base Model | 6.82 | 5.26 | 0.72 | 803 | 1303 |
| tulu-2-dpo-7b | 9.20 | 8.20 | 0.87 | 805 | 1663 |
| DPO ($\lambda = 1.0$) | 8.84 | 6.41 | 0.79 | 803 | 1230 |
| A-DPO ($\lambda = 0.0$) | 6.75 | 5.30 | 0.73 | 802 | 1287 |
| Multi-DPO ($\lambda = 0.5$) | 7.69 | 5.73 | 0.74 | 801 | 1208 |
| DPOP ($\lambda = 1.0$) | 8.51 | 6.23 | 0.79 | 802 | 1242 |
| A-DPOP ($\lambda = 0.0$) | 6.48 | 5.14 | 0.71 | 802 | 1318 |
| Multi-DPOP ($\lambda = 0.5$) | 8.26 | 6.17 | 0.77 | 802 | 1244 |

Our experiments have shown that the abductive preference learning method has made great improvements in recognizing the differences between the slight modification of image patterns. Notably, our accuracy is comparable with the accuracy gained by human evaluators as illustrated in Jain et al. (2024).

Table 6: Pairwise accuracy comparing base model, preference learning, abductive preference learning, and the multitask preference learning methods that combine both.

| Model | Accuracy |
|---|---|
| Qwen2.5-VL-3B | 50.0% |
| Qwen2.5-VL-7B | 40.7% |
| LLaVA1.5-7B-HF | 42.3% |
| Qwen2.5-VL-3B (A-DPO) | 87.0% |

Table 7: Evaluation Results on HUMORDB.

## 4.2 ABLATION STUDY: SIMILAR LEARNING DYNAMICS FOR ABDUCTIVE PREFERENCE LEARNING AND ORIGINAL PREFERENCE LEARNING

**Effect of $\lambda$ in multitask learning.** Recall the parameter $\lambda$ balancing the roles of preference learning and its abductive variants as illustrated in equation 3.2. To present the effect of different $\lambda$s on the evaluation performance of the trained model regarding the original preference learning's goal (Accuracy) and the goal of abductive preference learning (Abductive Accuracy), we do ablation studies with $\lambda \in \{0, 0.1, \ldots, 0.9, 1\}$ as reflected in Fig. 3. It can be observed that when $\lambda$ is ranging from 0.2 to 0.7, their performance on the evaluation set is similar. When $\lambda$ is greater than 0.7, putting more focus on the goal of the original preference learning, the Abductive Accuracy drops quickly. Similarly when $\lambda$ is smaller than 0.2. As a consequence, we consider $\lambda = 0.5$ a reasonable choice.

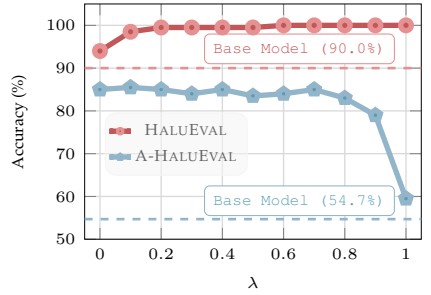
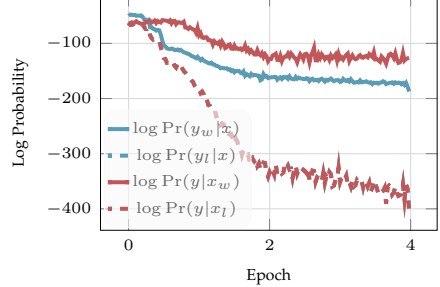

Figure 3: Ablation studies for the weight $\lambda$ of the original preference learning objective. Dashed lines indicate base model performance.

Figure 4: Training log-probabilities vs. epoch. Blue line imlies DPO training, while the red line implies A-DPO.

**Effect of log-probability gap ($\delta$, equation 3.1).** To explore the influence of the threshold $\delta$ on the generated dataset, we compare models trained on datasets constructed with $\delta = 1.0$ versus $= 0.1$, each for 4 epochs. Figure 5 plots abductive accuracy over epochs on two evaluation sets ($\delta = 0.1$ and $= 1.0$). Training on large gaps achieves near-perfect performance on the large-gap evaluation but fails to generalize to the small-gap evaluation ($\approx 40 - 48\%$ abductive accuracy). In contrast, training on small gaps yields high abductive accuracy on both evaluations ($\approx 85 - 91\%$ on 0.1 and $\approx 93 - 97\%$ on 1.0).

This indicates that exposure to small counterfactual margins during training is critical for robustness across margin regimes for A-DPO, whereas training solely on large margins leads to overspecialization, being identical to the pheonomenon of DPO as reported in Wu et al. (2024), which is also an motivation of online DPO as reported in Qi et al. (2024). This observation reveals that ADPO follows similar training patterns with DPO.

**Squeezing Effect in A-DPO.** We observed the squeezing effect of A-DPO during training, replicating the learning dynamics of DPO as reported in Ren & Sutherland (2024). More specifically, as illustrated in Fig. 4, we noticed that with more training epochs, both the log probability ratio of $\Pr(y \mid x_w)$ and $\Pr(y \mid x_l)$ decrease, which are the probabilities of selecting the hallucinated response based on the modified prompt and original prompt in our text only experiments, respectively. This is another validation of the fact that the abductive learning is showing similar training patterns as the original preference learning methods. Notably, both the $\Pr(y \mid x_w)$ and $\Pr(y_l \mid x)$ reflects the probability of choosing the hallucinated answer based on the original prompt in our dataset. These two curves overlap.

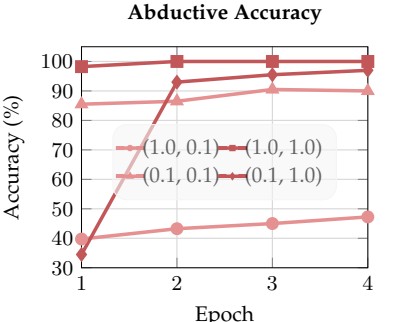
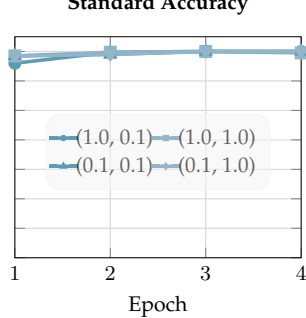

Figure 5: Effect of threshold $\delta$ on generalization. We employ $(\delta_t, \delta_e)$ to represent the evaluation performance on the dataset generated with $\delta_e$ of the model trained on dataset generated by $\delta_e$. Left: abductive accuracy on A-HALUEVAL (red). Right: standard accuracy (blue) on HALUEVAL. Each curve reports performance across 4 epochs; for the model trained with $\delta = 1.0$, checkpoints are averaged into two per epoch.

## 5 CONCLUSION

We introduced **abductive preference learning**, a paradigm that reverses the conditioning direction of standard preference optimization and is theoretically justified under mild assumptions. On datasets derived from HALUEVAL and ALPACAEVAL, abductive variants substantially improved prompt discrimination without sacrificing response alignment, and multitask training combined their complementary strengths. Finally, we demonstrated the applicability of this framework in multimodal settings such as sarcasm detection, highlighting its potential as a general tool for fine-grained alignment.

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

# A    APPENDIX

We first declare that we are using LLM only for the polishing of our manuscript.

In the following, we first present an ablation study for the text only abductive preference learning, adjusting the penalty parameter in Multi-DPOP. Afterwards, we present the proof for Proposition 2.1. Finally, we present the training hyperparameters we employed for our model training.

## A.1    ABLATION STUDY: MULTI-DPOP

From the ablation study of the penalty parameter from DPOP, we observed that the modification of $\lambda_{\text{DPOP}}$ is not altering the DPO accuracy. In contrast, the abductive accuracy is dropped with the increasing $\lambda_{\text{DPOP}}$. This is possibly led by the data generation framework of abductive HALUEVAL as illustrated in Table 3, that is, the pair $(x_l, y)$ on the abductive side is the same as the pair $(x, y_l)$ from the original perspective. In addition, the points are not dropping to a great extent, showing that the modification of parameter $\lambda_{\text{DPOP}}$ is not playing a critical role in the abductive learning framework.

Table 8: Ablation studies for the penalty $\lambda_{\text{DPOP}}$ of the original preference learning objective ($\lambda = 0.5$, (equation 3.2)).

| $\lambda_{\text{DPOP}}$ | Accuracy (HALUEVAL, %) | Accuracy (A-HALUEVAL, %) |
|---|---|---|
| 1.0 | 100.0 | 87.0 |
| 0.9 | 100.0 | 87.0 |
| 0.8 | 100.0 | 87.5 |
| 0.7 | 100.0 | 87.5 |
| 0.6 | 100.0 | 87.5 |
| 0.5 | 100.0 | 88.5 |
| 0.4 | 100.0 | 90.0 |
| 0.3 | 100.0 | 89.5 |
| 0.2 | 100.0 | 90.0 |
| 0.1 | 100.0 | 91.0 |
| 0.0 | 100.0 | 91.0 |

## A.2    PROOF OF PROPOSITION 2.1

Recall the formulation of abductive policy as formulated in equation 2.4. Under the assumption that the marginal distributions of prompts remain the same for both $\pi_\theta$ and $\pi_{\text{ref}}$, we have

$$\widetilde{\pi}_\theta(x \mid y) := \frac{\pi_\theta(y \mid x)p(x)}{q_\theta(y)}, \quad \widetilde{\pi}_{\text{ref}}(x \mid y) := \frac{\pi_\theta(y \mid x)p(x)}{q_{\text{ref}}(y)}.$$

Rewrite the learning objective function of A-DPO as follows:

$$\mathcal{L}_{\text{A-DPO}}(\pi_\theta; \pi_{\text{ref}})$$
$$= -\mathbb{E}_{(x_w, x_l, y) \sim \mathcal{D}} \left[ \log \sigma \left( \beta \log \frac{\widetilde{\pi}_\theta(x_w \mid y)}{\widetilde{\pi}_{\text{ref}}(x_w \mid y)} - \beta \log \frac{\widetilde{\pi}_\theta(x_l \mid y)}{\widetilde{\pi}_{\text{ref}}(x_l \mid y)} \right) \right]$$
$$= -\mathbb{E}_{(x_w, x_l, y) \sim \mathcal{D}} \left[ \log \sigma \left( \beta \log \frac{\pi_{\text{ref}}(y \mid x_w)p(x_w)/q_{\text{ref}}(y)}{\pi_\theta(y \mid x_w)p(x_w)/q_\theta(y)} - \beta \log \frac{\pi_{\text{ref}}(y \mid x_l)p(x_l)/q_{\text{ref}}(y)}{\pi_\theta(y \mid x_l)p(x_l)/q_\theta(y)} \right) \right]$$
$$= -\mathbb{E}_{(x_w, x_l, y) \sim \mathcal{D}} \left[ \log \sigma \left( \beta \log \frac{\pi_{\text{ref}}(y \mid x_w)}{\pi_\theta(y \mid x_w)} - \beta \log \frac{\pi_{\text{ref}}}{\pi_\theta(y \mid x_l)} \right) \right].$$

## A.3    TRAINING HYPERPARAMETERS

To begin with, our training setup is supported by 8-H100 GPUs.

### A.3.1    TEXT MODEL

- max_prompt_length: 1024;

- max_length: 2048;
- per_device_train_batch_size: 1;
- num_train_epochs: 5;
- learning_rate: 5e-7;
- gradient_accumulation_steps: 8;
- max_grad_norm: 1.0;
- lr_scheduler_type: "constant_with_warmup";
- warmup_ratio: 0.1;
- optimizer:
    - type: "adam";
    - weight_decay: "0.01";
    - betas: [0.9, 0.999];
    - eps: 1e-8.

### A.3.2 MULTIMODAL MODEL

- max_prompt_length: 8000;
- max_length: 8192;
- per_device_train_batch_size: 1;
- num_train_epochs: 5;
- learning_rate: 2e-6;
- gradient_accumulation_steps: 16;
- lr_scheduler_type: "constant_with_warmup";
- warmup_ratio: 0.2.

