# OpenReview forum: "Abductive Preference Learning"
_ICLR.cc/2026/Conference — Submitted to ICLR 2026_

### Official Review · Reviewer_AADk · 2025-10-23

**Soundness:** 2
**Presentation:** 3
**Contribution:** 1
**Rating:** 2
**Confidence:** 3

**Summary:**

This paper introduces **Abductive Preference Learning**,  a method that reverses the roles of prompts and responses in Direct Preference Optimization (DPO). The authors claim this improves model sensitivity to prompt variations and mitigates overconfidence in large language models. The authors demonstrate their method through experiments on several datasets, including HALUEVAL and AlpacaEval, and claim improvements in both response selection and prompt discrimination. The paper also extends the idea to multimodal settings with a humor detection experiment using the HUMORDB dataset. The main contribution of this paper is the introduction of a reversed conditioning method, which is validated theoretically and empirically.

**Strengths:**

1. The method is simple and easy to implement.
2. The paper is well-structured and covers both textual and multimodal scenarios.

**Weaknesses:**

1. **Weak Motivation:** The paper fails to formally define "prompt sensitivity" or establish its causal relationship to alignment goals and does not convincingly demonstrate the real-world relevance of the proposed method. While improvements in prompt discrimination are highlighted, the practical benefits of such improvements remain unclear.
2. **Overstated Novelty:** This paper presents a straightforward role-reversal of prompts and responses in DPO as its central contribution, but there is no new technics, no novel loss function, and no principled insight beyond swapping variables in an existing objective.
3. **Overstated Theoretical Contribution:** The theoretical justification relies on a trivial reordering of variables via Bayes' theorem. Proposition 2.1 is a rederivation of the original DPO result, offering no new probabilistic framework or novel insight.

**Questions:**

1. **Practical Significance:** What is the real-world impact of this technique? While the experiments show some improvement, there is insufficient evidence to demonstrate that these improvements would lead to significant practical benefits in broader applications.

2. **Root Cause:** The paper attributes the "overconfidence" issue in models to limitations in the training framework, but could this problem be due to poor prompt design rather than issues with the training framework itself? The root cause might be the prompts themselves rather than the optimization method.

If the authors can provide clearer evidence of the real-world impact of their method and address whether prompt design is the root cause of the overconfidence issue, I would be willing to increase the score.

---

### Official Review · Reviewer_BNbB · 2025-10-28

**Soundness:** 2
**Presentation:** 2
**Contribution:** 2
**Rating:** 2
**Confidence:** 4

**Summary:**

This paper proposes a fine-tuning paradigm that reverses the conventional preference optimization. Instead of training to prefer better responses for a fixed prompt (RLHF, DPO), they train them to prefer better prompts for a fixed response. They claim this inversion encourages sensitivity to subtle prompt differences, which in turn addresses overconfidence where the models give identical answers to modified inputs.

**Strengths:**

The paper introduces a somewhat novel conceptual framing, reversing the conditioning in preference learning to encourage prompt sensitivity. The writing is mostly clear. Empirically the authors demonstrate some modest improvements on standard benchmarks.

**Weaknesses:**

Sec 2.2 and Eq 2.3 is simply the standard preference loss applied to the prompt pair instead of response pair. As such, it is not a fundamentally new optimization. The authors do not show that this inversion yields different learning dynamics except when combined with the specifically curated data.

While the method improves "prompt discrimination", the practical utility of this metric is questionable. Real users rarely face paired prompts for the same answer. The small improvements on Alpacaeval further underscores that the proposed approceh has limited effect on general alignment and performance

**Questions:**

1. In the multimodal section, you attach the same question "is this image funny?". It would be informative to compare the proposed approach to a simple supervised baseline that directly trains the model to output Yes/No given the image?
2. In the text task, could you compare with a data-augmentation or contrastive training methods which would also encourage prompt sensitivity without reversing the conditioning

---

### Official Review · Reviewer_LJZx · 2025-11-01

**Soundness:** 3
**Presentation:** 3
**Contribution:** 3
**Rating:** 6
**Confidence:** 3

**Summary:**

This paper presents an approach to overcome over-confidence in large language models, where models are sure of their answer and do change their answer despite subtle differences in their prompt that ought to elicit a different response. The core of the idea is to switch from learning a preferred response from a pair of responses to a prompt, to instead learn a preferred prompt from a pair of prompts given a response. This way subtle but relevant changes can be applied to prompts so that the model can learn from these counterfactual examples. The results suggest that models learn from these adapted prompts in ways that are effective.

**Strengths:**

- Although I question the example of overconfidence in the provided example, it is clear that models are overconfident. This paper offers a relatively small change from the typical RLHF scenario that appears to be effective.
- The proposed switch from conditioning learning from pairs of prompts to pairs of queries allows specific and nuanced information that is relevant to driving preferences to be learned.

**Weaknesses:**

- The scope of the learning is somewhat limited in terms of the model (sizes/families). The results though are positive and the approach looks to be effective.

**Questions:**

- In the example that the model answers “no” that potato chips should not be eaten after being left out all night. Why is this overconfidence? I’m not sure that we can glean anything about confidence from the answer alone. There could be other underlying reasons, like hygiene, beyond the need to refrigerate. In fact, ChatGPT will now indicate that they likely will be stale and suggests how to “revitalize”.
- Line 026: A space is missing between the period and “Finally”.
- Line 095: explain that the blue variant is the original and red variant is the adapted information in the prompt.
- Line 240: “question, answer, hallucinated answer” > “question, hallucinated answer”.
- Line 350: The reference to Table 3 is wrong, it should be Table 4.
- Line 354: “A-DPO do little improvements” > “A-DPO does little to improve”.
- Line 357: “results can be when” — there is a word missing here?
- Line 359: “A-DPOP training is not dropping the” > “A-DPOP training does not drop the”.
- Line 361: “learning make improvements" > “learning makes improvements".
- Line 363: “Multi-DPOP are making improvements” > “Multi-DPOP make improvements”.
- I am unsure what to glean from Figure 5, and I do not find the caption helpful. I wonder if this could be clarified?

---

### Official Review · Reviewer_VhuU · 2025-11-01

**Soundness:** 2
**Presentation:** 2
**Contribution:** 1
**Rating:** 2
**Confidence:** 4

**Summary:**

This paper addresses the problem of LLM overconfidence, where models produce identical responses to prompts that differ in subtle but important ways (e.g., "Can I eat food left out overnight?" vs. "Can I eat potato chips left out overnight?"). The authors propose abductive preference learning, which reverses the conditioning direction in preference optimization by learning Pr(x|y) (which prompt is better for a given response) in addition to standard Pr(y|x) (which response is better for a given prompt). The method applies Bayes' theorem to swap roles of prompts and responses in DPO and DPOP objectives. Experiments use a dataset of 1,001 examples derived from HaluEval QA by modifying background knowledge. On the base model Tulu-2-7B, multitask DPOP improves response selection accuracy from 90% to 99.5% and prompt discrimination accuracy from 54.7% to 85%. Multimodal experiments on HumorDB show accuracy improvements from 50% to 87% for distinguishing humorous vs. non-humorous images.

**Strengths:**

**Interesting framework**: Proposition 2.1 shows that under the assumption of prompt distribution independence from model policies, the abductive DPO objective can be implemented by simply swapping (x,y) roles in the standard DPO loss. The derivation via Bayes' theorem is straightforward and the mathematical framework extends naturally to other preference learning methods (DPOP, GPO).

**Multimodal validation**: Extending the framework to vision-language models on HumorDB demonstrates generality beyond text. The problem setting (distinguishing minimally contrastive humorous/non-humorous image pairs) is a natural fit for sensitivity-focused learning.

**Weaknesses:**

**Severely undermotivated problem**: The paper never establishes that prompt insensitivity is a significant practical issue. The examples in Table 1 are cherry-picked edge cases—no evidence shows this is widespread. The claim that "GPT-5 and Claude Sonnet remain prone to overconfidence" is unsupported by any systematic evaluation. If this were a major problem affecting frontier models, there would be prior work addressing it. The absence of such work suggests the problem is either not important or easily solved by existing methods. The paper provides no error analysis of real model deployments, no user studies showing this causes failures, and no benchmark showing current models struggle with counterfactual sensitivity.

**Misleading framing as "overconfidence"**: The examples demonstrate insensitivity to prompt changes, not overconfidence. Overconfidence refers to miscalibrated probability estimates (assigning high confidence to incorrect predictions). Producing "Yes" to different questions is prompt-invariance or overgeneralization, not overconfidence. The paper conflates these distinct phenomena throughout, citing calibration papers (Leng et al., Xiao et al.) that address probability calibration, not prompt sensitivity. This conceptual confusion undermines the motivation.


**Weak evaluation scope**: The text evaluation is essentially limited to one task—HaluEval QA, a question-answering benchmark with factual answers. The paper never tests on instruction following (e.g., Alpaca), dialogue (e.g., MT-Bench), coding (e.g., HumanEval), reasoning (e.g., GPQA), or any of the diverse capabilities that define modern LLMs.

**Questions:**

1. Can you provide systematic evidence (not experiential evidence) that prompt insensitivity is a widespread problem in frontier models? Please report error rates on a benchmark specifically designed to test counterfactual sensitivity, or provide user studies showing this causes failures in practice. The cherry-picked examples in Table 1 are insufficient motivation.

---

### Meta-Review · Area_Chair_xHdc · 2025-12-30

**Summary:**

The primary weaknesses of this submission are its insufficient and sometimes misleading motivation, limited novelty, and weak empirical grounding. The paper frames the problem as "LLM overconfidence," but the presented examples and experiments instead reflect prompt insensitivity or overgeneralization, with no systematic evidence that miscalibrated confidence is involved or that this issue is widespread in modern frontier models. The claimed theoretical contribution is largely a trivial reparameterization of existing DPO-style objectives via Bayes’ rule, offering little new insight beyond swapping prompt–response roles, and the novelty is consequently overstated. Empirically, evaluation is narrow and heavily curated, focusing on a small modified HaluEval dataset and a single multimodal humor task, without demonstrating benefits on diverse, realistic LLM settings such as instruction following, dialogue, reasoning, or coding, nor against simpler baselines like data augmentation or contrastive training. Improvements are concentrated on "prompt discrimination," a metric whose practical relevance is unclear, and gains on broader alignment or utility benchmarks are modest.

**Reviewer Concerns:**

The authors did not provide any rebuttals.

**Reviewer Scores:**

The authors did not provide any rebuttals.

---

### Decision · Program_Chairs · 2026-01-26

Reject